# Comparing effectiveness of high-dose Atorvastatin and Rosuvastatin among patients undergone Percutaneous Coronary Interventions: A non-concurrent cohort study in India

**Debabrata Roy**[1]*, **Tanmay Mahapatra**[2], **Kaushik Manna**[1], **Ayan Kar**[1], **Md Saiyed Rana**[1], **Abhishek Roy**[1], **Pallab Kumar Bose**[1], **Barnali Banerjee**[2], **Srutarshi Paul**[2], **Sandipta Chakraborty**[2]

1 Department of Cardiology, Narayana Hrudayalaya Rabindranath Tagore International Institute of Cardiac Sciences, Kolkata, West Bengal, India, 2 Mission Arogya Health and Information Technology Research Foundation, Kolkata, West Bengal, India

* debroy67@ymail.com

## Abstract

### Introduction

Atorvastatin-80mg/day and Rosuvastatin-40mg/day are the commonest high-dose statin (3-hydroxy-3-methylglutaryl coenzyme-A reductase inhibitors) regimes for post-PCI (Percutaneous Coronary Interventions) patients to lower (by ≥50%) blood low-density-lipoprotein cholesterol (LDL-C). Dearth of conclusive evidence from developing world, regarding overall safety, tolerability and comparative effectiveness (outcome/safety/tolerability/endothelial inflammation control) of Rosuvastatin over Atorvastatin in high-dose, given its higher cost, called for an overall and comparative assessment among post-PCI patients in a tertiary cardiac-care hospital of Kolkata, India.

### Methods

A record-based non-concurrent cohort study was conducted involving 942 post-PCI patients, aged 18–75 years, on high-dose statin for three months and followed up for ≥one year. Those on Atorvastatin-80mg (n = 321) and Rosuvastatin-40mg (n = 621) were compared regarding outcome (death/non-fatal myocardial infarction: MI/repeated hospitalization/target-vessel revascularisation/control of LDL and high-sensitivity C-reactive protein: hsCRP), safety (transaminitis/myopathy/myalgia/myositis/rhabdomyolysis), tolerability (gastroesophageal reflux disease: GERD/gastritis) and inflammation control adjusting for socio-demographics, tobacco-use, medications and comorbidities using SAS-9.4.

### Results

Groups varied minimally regarding distribution of age/gender/tobacco-use/medication/comorbidity/baseline (pre-PCI) LDL and hs-CRP level. During one-year post-PCI follow up,

**Data Availability Statement:** All relevant data are within the paper and its Supporting Information files

**Funding:** The author(s) received no specific funding for this work.

**Competing interests:** The authors have declared that no competing interests exist.

none died. One acute MI and two target vessel revascularizations occurred per group. Repeated hospitalization for angina/stroke was 2.18% in Atorvastatin group vs. 2.90% in Rosuvastatin group. At three-months follow up, GERD/Gastritis (2.18% vs 4.83%), uncontrolled hs-CRP (22.74% vs 31.08%) and overall non-tolerability (4.67% vs. 8.21%) were lower for Atorvastatin group. Multiple logistic regression did show that compared to Atorvastatin-80mg, Rosuvastatin-40mg regime had poorer control of hs-CRP ($A_3OR = 1.45, p = 0.0202$), higher ($A_3OR = 2.07$) adverse effects, poorer safety profile ($A_3OR = 1.23$), higher GERD/Gastritis ($A_3OR = 1.50$) and poorer overall tolerability ($A_3OR = 1.50$).

## Conclusion

Post-PCI high dose statins were effective, safe and well-tolerated. High dose Rosuvastatin as compared to high dose Atorvastatin were similar in their clinical efficacy. Patients treated with Atrovastatin had significantly lower number of patients with hs-CRP (high-sensitivity C-reactive protein)/C-reactive protein (CRP) level beyond comparable safe limit and relatively better tolerated as opposed to Rosuvastatin-40mg. Thus given the lower price, Atorvastatin 80mg/day appeared to be more cost-effective. A head-to-head cost-effectiveness as well as efficacy trial may be the need of the hour.

## Introduction

Cardiovascular disease (CVD), a major killer, accounting for about 30% of current annual global deaths, has a well-known association with dyslipidemia [1–4]. Thus, in cardiovascular disease management, lipid lowering agents are considered important. Among them, 3-hydroxy-3-methylglutaryl coenzyme-A (HMG-CoA) reductase inhibitors or 'statins' are used widely as potent pharmacological choices for lowering low-density lipoprotein cholesterol (LDL-C) in blood [4–9]. Numerous studies tried to resolve the debates regarding the dosage of these statins for different indications. Having slight structural and pharmacokinetic differences, Atorvastatin and Rosuvastatin are the two most commonly used, compared and well-studied statins. Atorvastatin 80 mg/day or Rosuvastatin 40 mg/day are defined as high dose of statin because those average daily dosages reduce plasma LDL-C levels by 50% or greater [10]. Compared to moderate/low dose, statins in high dose can elicit better cardiovascular outcome among established CVD cases [11–16]. Efficacy/safety trials with high dose Atorvastatin such as AVERT, TNT, IDEAL, MIRACL found 15% to 36% reduction in the primary endpoints (cardiac death/myocardial infarction/ rehospitalization due to stroke/unstable angina/revascularization) [13, 15, 17, 18]. Trials like GISSI-HF, JUPITER on the other hand observed a marked reduction in the cardiac endpoints and overall better outcome with rosuvastatin [19–24]. Although in comparative low to moderate dose range, favourable outcomes were observed with Rosuvastatin, with high dose the efficacy of both were comparable [25, 26]. Similarly the safety and tolerability of both were analogous [27–31]. But comparative analysis between Rosuvastatin and Atorvastatin over 20 years treatment effect in a simulated trial using the Archimedes model and involving the results of CARDS, ASCOT, JUPITER, and the TNT trial revealed better potency of Rosuvastatin [32]. Despite these evidences, investigation on comparable safety or tolerability of high dose Rosuvastatin (40mg) vs. Atorvastatin (80mg) among Indians was unavailable.

In addition to the lack of confirmatory evidence regarding comparative efficacy of Rosuvastatin 40mg/day as opposed to Atorvastatin 80mg/day, owing to lack of head-to-head trials and conflicting results, the relative performance of these statins at high dose in endothelial inflammation control remained a unestablished. The common use of plasma level of highly sensitive C-reactive protein as an inflammatory marker, could test this, keeping the pleotropic effect of statins in mind.

Cost is another major factor that comes into play with statin therapy in developing nations. Availability of Atorvastatin in the generic forms provides a cheaper alternative to the high priced Rosuvastatin. In a low to moderate income setting, this difference is critical, as higher cost might ensue low adherence and even unreported discontinuation. In such a case if found to have similar effectiveness, Atorvastatin could be a more cost effective choice. There remained a paucity of information regarding the comparative effectiveness of the two high-dose statins on cardiovascular outcomes in developing country setting. Our study intends to compare the effectiveness in terms of outcome, safety and tolerability of both the statins at equivalent doses among post-PCI patients.

A record-based non-concurrent cohort study was thus contemplated to assess the safety and tolerability of high-dose statins and to compare the primary outcomes, safety and tolerability between Atorvastatin 80mg and Rosuvastatin 40mg among post-PCI patients among whom high dose statin is the current standard treatment of choice.

## Material and methods

### Ethical aspects

The study protocol and procedures were reviewed and approved (Ref. No.: NHRTIICS-EC/AP-2017) by the Ethics Committee of Rabindranath Tagore International Institute of Cardiac Sciences (RTIICS), Kolkata. Written informed consents were obtained from the patients, who were enrolled for the study.

### Eligibility criteria

Adult (age $\geq$ 18 years) patients with a final diagnosis of acute coronary syndrome [ACS which included: unstable angina (UA), non-ST-segment (NSTEMI) and ST-segment elevation myocardial infarction (STEMI)] [33, 34] or CCS (Canadian Cardiovascular Society) class III-IV chronic stable angina (unrelieved despite optimal medical therapy), who underwent Percutaneous Coronary Intervention (PCI) in Rabindranath Tagore

International Institute of Cardiac Sciences (RTIICS), Kolkata between 2009 and 2016 either electively or on emergency basis, discharged from the hospital with Atorvastatin 80 mg or Rosuvastatin 40 mg daily and unless reached any primary endpoint, underwent follow up for at least one year, were eligible for the study.

Patients were excluded if they were aged more than 75 years, on drugs which were known to induce or inhibit liver enzymes (rifampicin, ketoconazole etc.), known to be allergic/intolerant to statins, had a chronic kidney disease (CKD) stage $\geq$3 or LVEF $\leq$ 30% or any previous existing myopathy or neuro-deficit. Irrespective of baseline lipid profile, along with other standard medical therapy, the dosage of statin was continued for a post PCI period of three months followed by dose reduction to 10 mg/day respectively for Atorvastatin or Rosuvastatin, without any cross over.

## Covariates

From medical records, anonymous information regarding socio-demographic factors (age, gender), tobacco use [current/ex(non-user in last 3 months)/never users], other medications (antiplatelet, beta -blocker, angiotensin conversion enzyme inhibitors, angiotensin receptor blocker, calcium channel blocker, diuretic, nitrate, amiodarone etc.) prescribed post-stenting, the type of stent used during PCI (bare metal stent/drug eluting stents/mixed), existing medical history/comorbidities [diabetes mellitus, hypertension, hyperlipidemia, history of myocardial infarction, low left ventricular ejection fraction (LVEF%, ≤50/>50) [35] and history of prior PCI] were extracted. Composite indexes for Comorbidities and Medications were developed for each patient by determining their cumulative presence and requirements respectively.

## Outcome measures

The primary endpoints were determined by major adverse cardiac events (MACE) like death, non-fatal myocardial infarction [1], repeated hospital admission for angina/stroke and target vessel revascularisation during one year after PCI [36, 37]. Composite Primary Outcome Index was generated by weighted (weight for death > non-fatal MI > repeated hospital admission for angina/stroke or target vessel revascularisation) sum of the outcome (zero if nothing happened) components. Overall outcome was defined as satisfactory if composite outcome index was zero and unsatisfactory if it was more than zero.

Secondary outcome was measured by the magnitude of reduction in serum levels of Low-density lipoprotein (LDL) and high sensitivity C-reactive protein (hsCRP) at three months after PCI on continued high-dose statin treatment [38, 39]. Satisfactory reduction (adequate control/not beyond comparable safe limit) in serum LDL level was considered at three months post-PCI follow-up, either if the absolute serum LDL level was reduced to ≤70mg/dl or a reduction of serum LDL level by 50 percent over baseline (pre-PCI level) upon continued high-dose statin treatment [40]. Similarly for hs-CRP values, if the absolute value reduction was greater than 50% of the base line (pre-PCI plasma hs-CRP level) then satisfactory reduction (adequate control/not beyond comparable safe limit) was assumed [41]. Overall secondary outcome index was determined by sum of scores (zero for satisfactory/adequate and one for the opposite) for reduction/control of serum LDL and hs-CRP at three months after PCI. Patients with zero overall secondary outcome index was categorized as having overall satisfactory secondary outcome and unsatisfactory if the index was more than zero.

## Measures of safety and tolerability

Safety for high-dose statin therapy among post-PCI patients was assessed during first three post-PCI months, keeping in mind the most severe potential adverse effect of high-dose statin therapy in its various forms that could include transaminitis, myopathy, myalgia, myositis and rhabdomyolysis. A stringent safety cut-off level was defined by either any sign of myalgia or development of any condition culminating into dose-reduction or withdrawal of statin or ≥3-fold rise (beyond comparable safe limit) in creatine kinase (CK)/creatine phosphokinase (CPK) [42]/serum glutamic oxalo-acetic transaminase (SGOT)/ glutamic pyruvic transaminase (SGPT) [43] serum levels above the upper limit of normal. High-dose statin therapy with respective drugs (Atorvastatin/Rosuvastatin) was considered to be unsafe for patients experiencing any of the above while for the rest it was considered safe.

To measure the tolerability of the statins at high dose during these three months, signs of gastroesophageal reflux disease (GERD)/gastritis [44] were additionally considered with all the

components of safety. An Overall Intolerability Index was generated with subsequent tolerability categorization [good (well-tolerated)/poor (poorly tolerated)] in the similar fashion.

## Statistical analysis

Descriptive analysis [means (with corresponding 95% confidence intervals = 95%CI) for continuous and frequency with proportions expressed in percentages (along with corresponding 95%CIs)] was conducted to determine the distribution of socio demographic characteristics, tobacco use, stent type, other medications, clinical profile, outcome of PCI, along with safety and tolerability of high-dose statin therapy across two regimes (Atorvastatin 80mg vs Rosuvastatin 40mg). Comparability of the distributions across the regimes were determined by assessing the overlap between corresponding 95%CIs.

Multiple logistic regression analyses were conducted to determine whether compared to Atorvastatin 80mg, Rosuvastatin 40mg regime was associated [adjusted odds ratio (AOR) with corresponding 95%CI and p value] with statistically different outcome of PCI, as well as safety or tolerability of high-dose statin. To determine this three separate multiple logistic regression models were used. In Model 1, age, gender, tobacco use and stent type were adjusted (AOR expressed as $A_1OR$) as covariates. In Model 2 (AOR expressed as $A_2OR$), Comorbidity Index and in Model 3 (AOR expressed as $A_3OR$), Comorbidity and Medication Index were adjusted additionally.

All statistical analyses were performed using SAS version 9.4.

## Result

To the best of our knowledge this is the first effort to publish a head-to head comparison of safety/tolerability/effectiveness between high dose statin regimes in Indian post-PCI patients. Among 942 eligible patients who had undergone PCI during 2009–15 in a tertiary cardiac care centre of Kolkata, 321 were on Atorvastatin 80 mg while 621 were taking Rosuvastatin 40mg for three months after PCI. Both the groups (on Atorvastatin 80mg and Rosuvastatin 40mg respectively) appeared more or less similar regarding the distribution of age (mean age of 58.11 vs 57.72 years), mean Medication Index (1.43 vs 1.57), mean Comorbidity Index (1.74 vs 1.67), baseline (pre-PCI) mean serum LDL (77.45 vs 79.41) and hs-CRP level (33.30 vs 27.02) giving an impression that the baseline characteristics of the potential covariates along with immediate post-PCI prescribed management were quite comparable (as evidenced by the overlapping 95%CIs). (Table 1)

Patients in both these groups were predominantly male (91.90% vs 88.08%), about one third were exposed to tobacco (30.84% vs 38.49%), majority had drug-eluting stents used during PCI (61.68% vs 55.56%), nearly 60% had hypertension (60.75% vs 57.81%), about 40% were suffering from diabetes (40.19% vs. 38.16%), hyperlipidaemia was almost universal (97.82% vs 92.75%) and more than a third (35.83% vs 36.55%) had low (≤50%) LVEF. Overlapping 95%CIs demonstrated that distribution of these variables also did not vary much across the statin regime groups. (Table 1)

During one year post-PCI follow up, none of the post-PCI patients died and only one acute MI occurred in each group of patients. Repeated hospitalization for angina/stroke during this period happened among 2.18% patients in Atorvastatin group as opposed to 2.90% in Rosuvastatin group while number of cases with post-PCI target vessel revascularization in this period was also equally small (only 2) in both groups. We only observed one case of myalgia in Rosuvastatin 80mg group during follow up after 3 months. There were no observed significant (based on overlapping 95% Cis) differences in gastritis or any other measures for tolerability. (Table 2)

**Table 1. Comparative distribution of the sociodemographic, behavioral and clinical factors among patients receiving high-dose of two statins after undergoing Percutaneous Coronary Interventions (PCI) in a tertiary cardiac care hospital of Kolkata, 2009–2016 (N = 942).**

| Continuous variables | | Post-stenting statin regime | | | |
|---|---|---|---|---|---|
| | | Atorvastatin 80 | | Rosuvastatin 40 | |
| | | n | Mean (95%CI) | n | Mean (95%CI) |
| Age | | 321 | 58.11 (57.06–59.15) | 621 | 57.72 (56.94–58.50) |
| Medication index | | 321 | 1.43 (1.32–1.54) | 621 | 1.57 (1.49–1.65) |
| Co-morbidity index | | 321 | 1.74 (1.62–1.85) | 621 | 1.67 (1.59–1.75) |
| Pre-stenting serum level of Low-density lipoprotein (LDL) | | 287 | 77.45 (71.96–82.93) | 545 | 79.41 (76.50–82.32) |
| Pre-stenting serum level of high-sensitivity C-reactive protein (hs-CRP) | | 317 | 33.30 (28.16–38.43) | 603 | 27.02 (23.69–30.35) |
| Categorical variables | | n | Percentage (95%CI) | n | Percentage (95%CI) |
| Gender | Female | 26 | 8.10 (5.10–11.10) | 74 | 11.92 (9.36–14.47) |
| | Male | 295 | 91.90 (88.90–94.90) | 547 | 88.08 (85.53–90.64) |
| Any form of tobacco use | Never | 222 | 69.16 (64.08–74.24) | 382 | 61.51 (57.68–65.35) |
| | Ex-user | 37 | 11.53 (8.01–15.04) | 88 | 14.17 (11.42–16.92) |
| | Current user | 62 | 19.31 (14.97–23.66) | 151 | 24.32 (20.93–27.70) |
| Stent Type | BMS/Bare metal stent | 58 | 18.07 (13.84–22.30) | 155 | 24.96 (21.55–28.37) |
| | Drug eluting stent | 198 | 61.68 (56.34–67.03) | 345 | 55.56 (51.64–59.47) |
| | Both | 65 | 20.25 (15.83–24.67) | 121 | 19.48 (16.36–22.61) |
| Post-stenting prescribed medication | Beta Blocker | 86 | 26.79 (21.92–31.66) | 204 | 32.85 (29.15–36.55) |
| | Angiotensin converting enzyme inhibitors | 100 | 31.15 (26.06–36.25) | 178 | 28.66 (25.10–32.23) |
| | Angiotensin receptor blockers | 70 | 21.81 (17.27–26.35) | 150 | 24.15 (20.78–27.53) |
| | Calcium channel blocker | 39 | 12.15 (8.56–15.74) | 59 | 9.50 (7.19–11.81) |
| | Diuretics | 80 | 24.92 (20.16–29.68) | 147 | 23.67 (20.32–27.02) |
| | Nitrates | 11 | 3.43 (1.43–5.43) | 20 | 3.22 (1.83–4.61) |
| | Amiodarone | 109 | 33.96 (28.75–39.16) | 258 | 41.55 (37.66–45.43) |
| Existing medical history of | Diabetes mellites | 129 | 40.19 (34.79–45.58) | 237 | 38.16 (34.33–42.00) |
| | Hypertension | 195 | 60.75 (55.38–66.12) | 359 | 57.81 (53.92–61.71) |
| | Hyperlipidemia | 314 | 97.82% (96.21–99.43) | 576 | 92.75% (90.71–94.80) |
| | Myocardial infarction (MI) | 101 | 31.46 (26.36–36.57) | 148 | 23.83 (20.47–27.19) |
| | Low left ventricular ejection fraction | 115 | 35.83 (30.55–41.10) | 227 | 36.55 (32.76–40.35) |
| | Prior Percutaneous Coronary Intervention | 11 | 3.43 (1.43–5.43) | 23 | 3.70 (2.21–5.19) |

N = Total number of patients studied.

n = Number of patients in each group (across the category of different variables).

95% CI = 95% Confidence Interval.

Thus, mean Composite Primary Outcome Index was 0.03 and 0.04 in Atorvastatin and Rosuvastatin groups respectively with overlapping 95%CIs and proportions having unsatisfactory Overall Primary Outcome at one year post-PCI follow up were merely 3% (3.12% vs. 3.38%) across groups. (Table 2)

Secondary outcome parameters were assessed during three post-PCI months. During this period, between the statin groups, there were not much differences (in terms of the overlapping of 95%CIs) in mean serum LDL (54.62 vs 53.99mg/dl) or hs-CRP (4.38 vs 2.80 mg/l) levels as well as in the proportions of patients having these post-PCI 3month follow up levels beyond comparable safe limits for LDL (18.38% vs. 14.01%). But proportion of patients with 3-month post-PCI hs-CRP level beyond comparable safe limit (inadequate control) was considerably low in the Atorvastatin 80mg treated group [22.74% (95%CI = 18.13–27.35)] compared to their Rosuvastatin treated counterparts [31.08% (95%CI = 27.43–34.73)]. (Table 2)

**Table 2. Comparative distribution of the sociodemographic, behavioral and clinical factors among patients receiving high-dose of two statins after undergoing Percutaneous Coronary Interventions (PCI) in a tertiary cardiac care hospital of Kolkata, 2009–2016 (N = 942).**

| Continuous variables | | Atorvastatin 80 (n = 321) | | Rosuvastatin 40 (n = 621) | |
|---|---|---|---|---|---|
| | | n | Mean (95%CI) | n | Mean (95%CI) |
| Primary Outcome (during 1yr post-PCI) | Composite Primary Outcome Index | 321 | 0.03 (0.01–0.06) | 621 | 0.04 (0.02–0.05) |
| Secondary outcome (during 3mths post-PCI) | Serum Low-density lipoprotein (LDL) level | 291 | 54.62 (52.35–56.89) | 556 | 53.99 (51.85–56.13) |
| | hs-CRP (high-sensitivity C-reactive protein) | 292 | 4.38 (1.48–7.28) | 558 | 2.80 (2.29–3.31) |
| | Composite Secondary Outcome Index | 321 | 0.41 (0.35–0.47) | 621 | 0.45 (0.41–0.50) |
| Safety (during 3mths post-PCI) | Serum glutamic oxalo-acetic transaminase (SGOT) level | 298 | 28.75 (27.25–30.25) | 600 | 31.01 (29.59–32.43) |
| | Serum glutamate-pyruvate transaminase (SGPT) | 298 | 34.77 (32.60–36.93) | 600 | 38.42 (36.57–40.28) |
| | Serum creatine phosphokinase (CPK) level | 294 | 114.12 (104.32–123.92) | 595 | 120.02 (99.56–140.49) |
| | Overall safety index | 321 | 0.03 (0.01–0.05) | 621 | 0.04 (0.02–0.06) |
| Tolerability | Overall intolerability index (during 3mths post-PCI) | 321 | 0.05 (0.03–0.08) | 621 | 0.09 (0.06–0.11) |
| **Categorical variables** | | **n** | **Percentage (95%CI)** | **n** | **Percentage (95%CI)** |
| Primary Outcome (during 1yr post-PCI) | Death | 0 | - | 0 | - |
| | Acute myocardial infarction (AMI) | 1 | 0.31 (0.00–0.92) | 1 | 0.16 (0.00–0.48) |
| | Repeated hospitalization for angina/stroke | 7 | 2.18 (0.57–3.79) | 18 | 2.90 (1.58–4.22) |
| | Target vessel revascularization | 2 | 0.62 (0.00–1.49) | 2 | 0.32 (0.00–0.77) |
| | Overall — Satisfactory | 311 | 96.88 (94.97–98.80) | 600 | 96.62 (95.19–98.04) |
| | Overall — Unsatisfactory | 10 | 3.12 (1.20–5.03) | 21 | 3.38 (1.96–4.81) |
| Secondary Outcome (during 3mths post-PCI) | Low-density lipoprotein (LDL) beyond comparable safe limit (inadequate control) | 59 | 18.38 (14.12–22.64) | 87 | 14.01 (11.27–16.75) |
| | hs-CRP (high-sensitivity C-reactive protein)/C-reactive protein (CRP) level beyond comparable safe limit (inadequate control) | 73 | 22.74 (18.13–27.35) | 193 | 31.08 (27.43–34.73) |
| | Overall — Satisfactory | 200 | 62.31 (56.98–67.64) | 364 | 58.62 (54.73–62.50) |
| | Overall — Unsatisfactory | 121 | 37.69 (32.36–43.02) | 257 | 41.38 (37.50–45.27) |
| Safety (during 3mths post-PCI) | Myalgia | 0 | 0.00 (0.00–0.00) | 1 | 0.16 (0.00–0.48) |
| | SGOT beyond comparable safe limit (inadequate control) | 1 | 0.31 (0.00–0.92) | 2 | 0.32 (0.00–0.77) |
| | SGPT beyond comparable safe limit (inadequate control) | 0 | 0.00 (0.00–0.00) | 2 | 0.32 (0.00–0.77) |
| | Overall Liver Function test beyond comparable safe limit (inadequate control) | 1 | 0.31 (0.00–0.92) | 3 | 0.48 (0.00–1.03) |
| | CPK beyond comparable safety limit (inadequate control) | 6 | 1.87 (0.38–3.36) | 8 | 1.29 (0.40–2.18) |
| | Discontinuation/reduction of statin dosage due to any adverse effects | 3 | 0.93 (0.00–1.99) | 13 | 2.09 (0.96–3.22) |
| | Overall safety — Safe | 313 | 97.51 (95.79–99.22) | 600 | 96.62 (95.19–98.04) |
| | Overall safety — Unsafe | 8 | 2.49 (0.78–4.21) | 21 | 3.38 (1.96–4.81) |

(*Continued*)

**Table 2.** (Continued)

| Continuous variables | | | Atorvastatin 80 (n = 321) | | Rosuvastatin 40 (n = 621) | |
|---|---|---|---|---|---|---|
| | | | n | Mean (95%CI) | n | Mean (95%CI) |
| Tolerability (at 3 month follow up) | GERD/Gastritis | No | 314 | 97.82 (96.21–99.43) | 591 | 95.17 (93.48–96.86) |
| | | Yes | 7 | 2.18 (0.57–3.79) | 30 | 4.83 (3.14–6.52) |
| | Overall tolerability | Good | 306 | 95.33 (93.01–97.65) | 570 | 91.79 (89.62–93.95) |
| | | Poor | 15 | 4.67 (2.35–6.99) | 51 | 8.21 (6.05–10.38) |

SGOT: Serum Glutamic Oxalo-acetic Transaminase.

SGPT: Serum glutamate-pyruvate transaminase.

CPK: Creatine phosphokinase.

Mean Composite Secondary Outcome Index was almost same for both the groups (0.41 vs 0.45) and close to 40% patients overall had some (37.69% vs. 41.38%) unsatisfactory secondary outcome. (Table 2)

Three months after the high-dose statin therapy across both regime groups, comparable SGOT, SGPT, CPK levels made the mean Overall Safety Index appear same (0.03 vs. 0.04). Proportion of patients for whom high-dose statin therapy found to be unsafe were also very low (2.49% vs. 3.38%) for both the drugs. Still it appeared that the mean serum levels of SGOT (28.75 vs. 31.01) and SGPT (34.77 vs. 38.02) were marginally lower (major portion of the 95% Cis did not overlap) for Atorvastatin compared to Rosuvastatin group. (Table 2)

Regarding tolerability it appeared that GERD/Gastritis was a bit less (through mostly non-overlapping 95%CI) common (2.18% vs 4.83%) in the Atorvastatin 80mg treated group, leading to a relatively lower mean Overall Intolerability Index (0.05 vs. 0.09) for Atorvastatin 80mg group. Proportion of patients in Atorvastatin 80mg group having poor Overall Tolerability (4.67% vs. 8.21%) was also almost half (95%CIs mostly non-overlapping) of that in the Rosuvastatin group. (Table 2)

Multiple logistic regression models (all three, with or without adjustment for Comorbidity and Medication Index) did show that with reference to Atorvastatin 80mg regime, patients on Rosuvastatin 40 mg were more likely ($A_3OR$ = 1.45, p = 0.0202) to have hs-CRP levels beyond

**Table 3. Association of Post PTCA Statin regime (Ref: Atorvastatin 80mg) with primary and secondary outcomes among patients receiving high-dose of two statins after undergoing Percutaneous Coronary Interventions (PCI) in a tertiary cardiac care hospital of Kolkata, 2009–2016 (N = 942).**

| | | Adjusted Model 1 | | Adjusted Model 2 | | Adjusted Model 3 | |
|---|---|---|---|---|---|---|---|
| | | $A_1OR$ (95%CI) | p value | $A_2OR$ (95%CI) | p value | $A_3OR$ (95%CI) | p value |
| **Primary outcome** | Repeated hospitalization for angina/stroke (Ref = No) | 1.30 (0.54–3.17) | 0.5586 | 1.20 (0.49–2.95) | 0.6854 | 1.33 (0.55–3.23) | 0.5317 |
| | Poor overall primary outcome (Ref = Good) | 1.06 (0.49–2.29) | 0.8811 | 0.96 (0.44–2.10) | 0.9258 | 1.08 (0.50–2.34) | 0.8445 |
| **Secondary outcome** | LDL level beyond comparable safe limit (adequate control) (Ref = No) | 0.72 (0.50–1.04) | 0.0790 | 0.71 (0.49–1.02) | 0.0664 | 0.73 (0.51–1.05) | 0.0918 |
| | CRP level beyond comparable safe limit (Ref = No) | **1.47 (1.07–2.01)** | **0.0175** | **1.46 (1.06–2.00)** | **0.0192** | **1.45 (1.06–2.00)** | **0.0202** |
| | Poor overall secondary outcome (Ref = Good) | 1.12 (0.85–1.49) | 0.4156 | 1.11 (0.84–1.47) | 0.4571 | 1.11 (0.84–1.47) | 0.4740 |

Model 1 adjusted for age, gender, tobacco use and stent type.

Model 2 additionally adjusted for Comorbidity Index.

Model 3 additionally adjusted for Medication Index.

comparable safe limits. Except this neither any other component nor the overall primary and secondary outcome of PCI did differ statistically across the two high-dose regimes of statins. (Table 3)

Relatively higher likelihood (A$_3$OR = 2.07) of having adverse effects needing dose reduction/discontinuation of statin, having relatively poorer safety profile (A$_3$OR = 1.23), suffering from GERD/Gastritis (A$_3$OR = 1.50) and having overall poor tolerability (A$_3$OR = 1.50) were apparent (as 95%CI limits were shifted more towards positive association) among patients on Rosuvastatin 40mg as opposed to their counterparts on Atorvastatin 80mg, although results were not statistically significant due to sparse data points in the category of patients for whom high-dose statins (either regime) appeared unsafe or poorly tolerated. (Table 4)

## Discussion

In this record-based non-concurrent cohort study involving 942 patients aged 18–75 years, who had undergone PCI in a tertiary cardiac care center in Kolkata, India, between 2009 and 2016, post-PCI regimes of high dose statins (321 on Atorvastatin 80mg and 621 on Rosuvastatin 40mg) were found to be quite safe (for 97%) and well tolerated (by 93%).

Baseline characteristics of the potential covariates, existing comorbidities and the immediate post-PCI prescribed management were quite similarly distributed among the study subjects across the statin regime groups (Atorvastatin 80mg and Rosuvastatin 40mg) making the two groups quite comparable.

Corroborating with prior studies [23, 45], based on primary end-points of MACE, PCI was found to be producing mostly (in 97%) satisfactory primary outcomes (similarly across statin regimes, as evidenced earlier also [46]) for all eligible post-PCI patients recruited in the study. During the one-year post-PCI follow up: nobody died in either group, only one patient in each group had acute MI, repeated hospitalization for angina/stroke were observed in only a small proportion and post-PCI target vessel revascularization was needed for only a few.

Secondary outcomes across statin regimes were compared using serum level of LDL as an indicator for hyperlipidaemia and hs-CRP as the marker for inflammation, three months after PCI. Regarding the control of serum LDL, both Atorvastatin 80mg and Rosuvastatin 40mg regimes appeared to be about equally effective in achieving overall satisfactory secondary outcomes in approximately two third of the patients during this follow up. Among these patients, majority had satisfactory control on serum lipids (84%) and plasma hs-CRP levels (72%) as also were evidenced previously by other researchers [47]. Based on previous studies it also appeared that cumulative effect of serum LDL levels and plasma CRP indicated important implications on atherosclerosis [48, 49].

But compared to those on Rosuvastatin 40mg, patients on Atorvastatin 80mg had statistically significantly better reduction in plasma levels of hs-CRP after adjustment for all other factors, indicating that this high-dose regime of Atorvastatin was more effective in inflammation control as opposed to the Rosuvastatin regime. Elsewhere also, the role of Atorvastatin in successfully reducing CRP level had been demonstrated previously [50, 51].

Both regimes of high dose statins were found to be quite safe (for 97%) for the post-PCI patients as during three post-PCI month follow up, very few patients (in both statin groups) developed deranged liver function, high serum CK/CPK level or development of any adverse events culminating into dose-reduction or withdrawal. Similar findings were also reported from the POLARIS study, STELLER study and others [52–56] Digging deeper, it was evident that relatively better safety profile may be assigned to high dose Atorvastatin (vs. Rosuvastatin) regime.

**Table 4. Association of Post PTCA Statin regime (Ref: Atorvastatin 80mg) with safety and tolerability among patients receiving high-dose of two statins after undergoing Percutaneous Coronary Interventions (PCI) in a tertiary cardiac care hospital of Kolkata, 2009–2016 (N = 942).**

| | | | Adjusted Model 1 | | Adjusted Model 2 | | Adjusted Model 3 | |
|---|---|---|---|---|---|---|---|---|
| | | | $A_1OR$ (95% CI) | p value | $A_2OR$ (95% CI) | p value | $A_3OR$ (95% CI) | p value |
| **Safety** | Any adverse effects which needed dose reduction or discontinuation of statins (Ref = No) | Yes | 2.16 (0.61–7.71) | 0.2338 | 2.20 (0.61–7.87) | 0.2274 | 2.07 (0.58–7.41) | 0.2617 |
| | Overall safety profile (Ref = Good) | Poor | 1.27 (0.55–2.91) | 0.5801 | 1.30 (0.56–3.02) | 0.5379 | 1.23 (0.53–2.83) | 0.6355 |
| **Tolerability** | Suffered from: GERD/Gastritis (Ref = No) | Yes | 2.09 (0.90–4.84) | 0.0846 | 1.96 (0.84–4.56) | 0.1171 | 2.16 (0.93–5.00) | 0.0728 |
| | Overall Tolerability (Ref = Good) | Poor | 1.69 (0.93–3.06) | 0.0869 | 1.63 (0.90–2.98) | 0.1091 | 1.69 (0.93–3.07) | 0.0854 |

Model 1 adjusted for age, gender, tobacco use and stent type.

Model 2 additionally adjusted for Comorbidity Index.

Model 3 additionally adjusted for Medication Index.

As also evidenced in available literature [49], high dose statins were mostly (93%) well tolerated by the patients recruited in the current study. GERD/Gastritis occurred a bit less commonly among patients on Atorvastatin 80 compared to their counterparts on Rosuvastatin 40mg. Overall tolerability also appeared to be shed better among those on high dose Atorvastatin compared to the other drug.

The current study also had some limitations. Unequal dose reduction may appear as a potential factor for variation in the outcome. Based on the available literature on therapeutic levels and dosage of Atorvastatin vs Rosuvastatin, comparison after dose reduction to 10mg for both was found to be most relevant, therapeutically indicated, plausible and thus conducted. As per the available evidences, to have equal dose reduction, comparison between reductions to Atorvastatin 10mg vs Rosuvastatin 5mg was not possible [57] to maintain the therapeutic target of lowering LDL level to ≤50% among post-PCI patients, lowered dose for both had to be 10mg [57, 58]. This dose being the most commonly used as well as median lowered dose of statin in such high-risk groups in real world scenario (especially among Asians) had to be administered and thus compared in this study [59–62]. Available literature strongly suggesting comparable outcome in terms of LDL lowering with Atorvastatin 10mg vs Rosuvastatin 10mg in real world actually made it more relevant [60]. Furthermore, the comparison of therapeutic impact of dose reduction remained beyond the pursuit of this effort.

Although, in Indian context, Rosuvastatin fared better regarding plaque regression and lipid control [63], with some more evidence elsewhere [63–65], specific data on such high-dose comparison in post-PCI were scanty. Despite the limitations, through parge sample size and robust analysis, the current could probably generate important insight regarding direct comparison of effectiveness in terms of controlling inflammation, preventing major adverse cardiac events and tolerance of high-dose statins among Indian post-PCI patients.

## Conclusion

High dose statins were found to be very effective, quite safe and well tolerated by post-PCI patients. Post-PCI regime of Atorvastatin 80mg/day was found to be more effective in controlling inflammatory process and relatively better tolerated as opposed to Rosuvastatin 40mg. While both these regimes were found to be equally effective in lowering serum LDL level and preventing major adverse cardiac events among post-PCI patients, given its relatively less

market price, Atorvastatin 80mg/day appeared to be a more cost-effective high-dose option as the lipid lowering regime to be prescribed immediate post-PCI. A head-to-head cost-effectiveness as well as efficacy trial may be the need of the hour.

## Supporting information

**S1 Data. Data description codebook.**
(XLSX)

**S2 Data. Dataset.**
(XLS)

## Acknowledgments

The authors want to express their sincere gratitude to all the colleagues who helped the processes starting from getting the data collected, collated, digitized, analysed and interpreted. The authors are specifically inclined to acknowledge the contribution of the Mission Arogya Health and Information Technology Research Foundation staff–Shankar Bhattacharya, Moloy Saha, Kaushik Ghosh, Partha Mukherjee, Arindum Kundu in this regard. Authors also acknowledge the support of the patients who participated in the study and contributed their time by providing the required information.

## Author Contributions

**Conceptualization:** Debabrata Roy.

**Data curation:** Debabrata Roy, Ayan Kar, Md Saiyed Rana, Abhishek Roy, Barnali Banerjee, Srutarshi Paul, Sandipta Chakraborty.

**Formal analysis:** Tanmay Mahapatra, Barnali Banerjee, Srutarshi Paul, Sandipta Chakraborty.

**Investigation:** Debabrata Roy, Kaushik Manna, Ayan Kar, Md Saiyed Rana, Abhishek Roy, Pallab Kumar Bose.

**Methodology:** Debabrata Roy, Tanmay Mahapatra, Kaushik Manna, Ayan Kar, Md Saiyed Rana, Abhishek Roy, Pallab Kumar Bose, Barnali Banerjee, Srutarshi Paul, Sandipta Chakraborty.

**Project administration:** Debabrata Roy, Kaushik Manna, Ayan Kar, Md Saiyed Rana, Abhishek Roy.

**Software:** Tanmay Mahapatra, Barnali Banerjee, Srutarshi Paul, Sandipta Chakraborty.

**Supervision:** Debabrata Roy, Md Saiyed Rana.

**Validation:** Debabrata Roy, Tanmay Mahapatra, Kaushik Manna, Ayan Kar, Pallab Kumar Bose, Barnali Banerjee, Sandipta Chakraborty.

**Visualization:** Barnali Banerjee.

**Writing – original draft:** Tanmay Mahapatra, Kaushik Manna, Ayan Kar, Md Saiyed Rana, Abhishek Roy, Pallab Kumar Bose, Srutarshi Paul, Sandipta Chakraborty.

**Writing – review & editing:** Debabrata Roy, Tanmay Mahapatra, Sandipta Chakraborty.

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
