## [Decision Letter · Decision Letter 0]

28 Feb 2020

PONE-D-19-32015

Comparing effectiveness of high-dose Atorvastatin and Rosuvastatin among patients undergone Percutaneous Coronary Interventions: a non-concurrent cohort study in India

PLOS ONE

Dear Dr. Roy,

Thank you for submitting your manuscript to PLOS ONE. After careful consideration, we feel that it has merit but does not fully meet PLOS ONE’s publication criteria as it currently stands. Therefore, we invite you to submit a revised version of the manuscript that addresses the points raised during the review process.

We would appreciate receiving your revised manuscript by Mar 30 2020 11:59PM. To enhance the reproducibility of your results, we recommend that if applicable you deposit your laboratory protocols in protocols.io, where a protocol can be assigned its own identifier (DOI) such that it can be cited independently in the future. For instructions see: http://journals.plos.org/plosone/s/submission-guidelines#loc-laboratory-protocols

We look forward to receiving your revised manuscript.

Kind regards,

Corstiaan den Uil

Academic Editor

PLOS ONE

Journal Requirements:

2. Please amend your list of authors on the manuscript to ensure that each author is linked to an affiliation. Authors’ affiliations should reflect the institution where the work was done (if authors moved subsequently, you can also list the new affiliation stating “current affiliation:….” as necessary).

Reviewers' comments:

Reviewer's Responses to Questions

**Comments to the Author**

1. Is the manuscript technically sound, and do the data support the conclusions?

Reviewer #1: Partly

Reviewer #2: Yes

2. Has the statistical analysis been performed appropriately and rigorously? 

Reviewer #1: Yes

Reviewer #2: Yes

3. Have the authors made all data underlying the findings in their manuscript fully available?

Reviewer #1: Yes

Reviewer #2: Yes

4. Is the manuscript presented in an intelligible fashion and written in standard English?

Reviewer #1: Yes

Reviewer #2: Yes

5. Review Comments to the Author

Reviewer #1: Please explain the equivalence between reduction to 10 mg in both arms from Rosuvastatin 40 mg vs Atorvastatin 80 mg after three months. Outcome may vary because of the unequal reduction of doses.

Kindly further explain the differences between the side effects in two groups like myalgia/statin intolerance.

Reviewer #2: The authors studied high dose Atorvastatin as compared to high dose Rosuvastatin post Percutaneous Coronary Interventions. This was an observation study with its limitations. The authors concluded that "Post-PCI regime of

Atorvastatin-80mg/day was more effective in controlling endothelial inflammation and relatively better tolerated as opposed to Rosuvastatin-40mg. Would would like to congratulate the authors for their work. I have the following observation to make.

Abstract:

The conclusion presented in the manuscript sounds very biased. High dose Rosuvastatin as compared to Atorvastatin reported a non significant difference in primary outcome, as suggested by overlapping confidence intervals. The decrease in hs-CRP (high-sensitivity C-reactive protein) is an indirect evidence of endothelial inflammation. Hence it would be better concluded as- " High dose Rosuvastatin as compared to high dose Atorvastatin are similar in their clinical efficacy. Patients treated with Atrovastatin had significantly lower number of patients with hs-CRP (high-sensitivity C-reactive protein)/C-reactive protein (CRP) level beyond comparable safe limit".

The authors should discuss why Atorvastatin was relatively better than Rosuvastatin, at least in Indian context. There is considerable evidence regarding Rosuvstatin being a better statin as compared to Atorvastatin, with regards to plaque regression and better lipid control. Anything that explains why this results were not reproduced in the present study.

The introduction is larger than discussion in the present manuscript. I would prefer seeing more content discussing the results in an original research submission. The introduction section can be shortened.

6. PLOS authors have the option to publish the peer review history of their article (what does this mean?). If published, this will include your full peer review and any attached files.

Reviewer #1: No

Reviewer #2: No

---

## [Author Response · Author response to Decision Letter 0]

28 Apr 2020

Response to Editor’s Comments

• A rebuttal letter that responds to each point raised by the academic editor and reviewer(s). This letter should be uploaded as separate file and labeled 'Response to Reviewers'.

• A marked-up copy of your manuscript that highlights changes made to the original version. This file should be uploaded as separate file and labeled 'Revised Manuscript with Track Changes'.

• An unmarked version of your revised paper without tracked changes. This file should be uploaded as separate file and labeled 'Manuscript'.

Done

Journal Requirements:

We have done the needful as instructed.

2. Please amend your list of authors on the manuscript to ensure that each author is linked to an affiliation. Authors’ affiliations should reflect the institution where the work was done (if authors moved subsequently, you can also list the new affiliation stating “current affiliation:….” as necessary).

This work was conducted in collaboration between two organizations: 

1. Department of Cardiology, Narayana Hrudayalaya Rabindranath Tagore International Institute of Cardiac Sciences, Kolkata, West Bengal, India

2. Mission Arogya Health and Information Technology Research Foundation, Kolkata, West Bengal, India

Doctors attached to the Department of Cardiology of Narayana Hrudayalaya Rabindranath Tagore International Institute of Cardiac Sciences, Kolkata, West Bengal, India: Dr. Debabrata Roy, Dr. Kaushik Manna, Dr. Ayan Kar, Dr. Saiyed Rana, Dr. Abhishek Roy, Dr. Pallab Bose and Epidemiologist/Public Health Specialists of Mission Arogya Health and Information Technology Research Foundation, Kolkata, West Bengal, India: Dr. Tanmay Mahapatra, Barnali Banerjee, Srutarshi Paul and Dr. Sandipta Chakraborty, collaborated together to conduct this research, analysis and manuscript development.

During submission of the manuscript, the affiliations were marked serially with numerical superscripts and while doing that mistakenly some of the authors were marked with 3 and 4 as affiliation marker. Actually, only two such affiliating organizations were to be listed. The mistakes have been corrected in the revised manuscript. We are sorry for the mistake. 

Review Comments to the Author

Reviewer #1: Please explain the equivalence between reduction to 10 mg in both arms from Rosuvastatin 40 mg vs Atorvastatin 80 mg after three months. Outcome may vary because of the unequal reduction of doses.

We are thankful to the reviewer for this valuable opinion and we are extremely sorry for not explaining this earlier in the previously submitted version of the manuscript. While we agree that unequal reduction of doses may apparently appear as a potential factor for variation in the outcome, but based on the available literature on therapeutic levels and dosage of Atorvastatin vs Rosuvastatin, comparison after dose reduction to 10mg for both was found to be most relevant, therapeutically indicated and plausible. The reasons being:

1. To have equal reduction of dose, we had the option of comparing between reductions to Atorvastatin 10mg vs Rosuvastatin 5mg. but it was not possible [1]because:

a. Therapeutic target for the management of patients who had undergone Percutaneous Coronary Intervention, was to maintain the statin dose in such level that LDL remained reduced by 50% or more.

b. Real world data shows that with Rosuvastatin 5mg, LDL was not expected to be reduced by more than 50%[1, 2]

2. The most commonly used regimes of statin in such high-risk groups in real world scenario remained to be 10mg for both Atorvastatin and Rosuvastatin with this being the median dose for both across such studies. [1] Furthermore this dose for both the drugs was the most common baseline statin therapy.[6]

3. Comparisons between 10mg dosing of Atorvastatin vs Rosuvastatin were suggested in literature, with aforementioned biological plausibility in studies conducted among Asian patients.[3]

4. Available literature also strongly suggested comparable outcome in terms of LDL lowering with Atorvastatin 10mg vs Rosuvastatin 10mg in real world.[4]

5. Studies that did compare baseline lower dose of these two statins also did compare between Atorvastatin 10mg vs Rosuvastatin 10mg before switching to higher doses.[1, 5]

Owing to these evidences, we did include patients in this study with dose reduction to 10mg for both as the literature mostly suggested comparability of therapeutic indications of Atorvastatin with Rosuvastatin respectively at 80mg vs 40mg at high dose and 10mg each at low dose.[1-4, 6] It can also be noted that comparison of therapeutic impact of such dose reduction was not made in this manuscript.

In the revised manuscript we have incorporated this limitation and changes are marked with tracking.

Kindly further explain the differences between the side effects in two groups like myalgia/statin intolerance.

We are thankful to the reviewer for this valuable opinion and we are extremely sorry for not explaining this earlier in the previously submitted version of the manuscript. In the revised manuscript we have incorporated this as per the instruction and changes are marked with tracking.

Reviewer #2: 

The authors studied high dose Atorvastatin as compared to high dose Rosuvastatin post Percutaneous Coronary Interventions. This was an observation study with its limitations. The authors concluded that "Post-PCI regime of

Atorvastatin-80mg/day was more effective in controlling endothelial inflammation and relatively better tolerated as opposed to Rosuvastatin-40mg. Would would like to congratulate the authors for their work. I have the following observation to make.

Abstract:

The conclusion presented in the manuscript sounds very biased. High dose Rosuvastatin as compared to Atorvastatin reported a non significant difference in primary outcome, as suggested by overlapping confidence intervals. The decrease in hs-CRP (high-sensitivity C-reactive protein) is an indirect evidence of endothelial inflammation. Hence it would be better concluded as- " High dose Rosuvastatin as compared to high dose Atorvastatin are similar in their clinical efficacy. Patients treated with Atrovastatin had significantly lower number of patients with hs-CRP (high-sensitivity C-reactive protein)/C-reactive protein (CRP) level beyond comparable safe limit".

We are very much thankful to the reviewer for the applause and not only pointing out the apparently undue emphasize in our conclusion, but also providing us with the corrected expression. We have incorporated this as per the instruction in the conclusion of the abstract and changes are marked with tracking.

The authors should discuss why Atorvastatin was relatively better than Rosuvastatin, at least in Indian context. There is considerable evidence regarding Rosuvstatin being a better statin as compared to Atorvastatin, with regards to plaque regression and better lipid control. Anything that explains why this results were not reproduced in the present study.

We are grateful to the reviewer, for pointing this out and we are extremely sorry for not discussing this contrast in our manuscript in the previously submitted version. 

We had discussed this in introduction, mentioning: “Trials like GISSI-HF, JUPITER on the other hand observed a marked reduction in the cardiac endpoints and overall better outcome with rosuvastatin treatment” and “But comparative analysis between Rosuvastatin and Atorvastatin over 20 years treatment effect in a simulated trial using the Archimedes model and involving the results of CARDS, ASCOT, JUPITER, and the TNT trial revealed better potency of Rosuvastatin”.

Actually, what we found was, while in Indian context, Rosuvastatin was found to do better with regards to plaque regression, better lipid control and some more, those comparisons were mainly made among patients with comparable dosing across whole spectrum of therapeutic indications.[7, 8] Also tolerability of these high dose statins remained a question for Indian patients, with literature suggesting that direct comparison with patients on similar indication, keeping the universe for the research for comparison limited to such specific high-risk groups were to be conducted to infer on those.[7-9] Comparison of Atorvastatin 80mg with Rosuvastatin 40mg among post-PCI patients was thus undertaken to compare their overall effectiveness in terms of Major Adverse Cardiac Event while tolerance and inflammation control were compared directly to address that specific research question.

In the revised manuscript we have incorporated these discussions to ensure that the points of view remain fully inclusive and rationale for the inference get further clarified.

The introduction is larger than discussion in the present manuscript. I would prefer seeing more content discussing the results in an original research submission. The introduction section can be shortened.

We are very much indebted for this valuable opinion of the reviewer and admit that, while summarizing background literature, we were unable to shorten the introduction further, which should have been done in addition to strengthening the discussion furthermore, contextualizing our results, in light of available 

References: 

1. Bullano MF, Kamat S, Wertz DA, Borok GM, Gandhi SK, McDonough KL, et al. Effectiveness of rosuvastatin versus atorvastatin in reducing lipid levels and achieving low-density-lipoprotein cholesterol goals in a usual care setting. Am J Health Syst Pharm. 2007;64(3):276-84. Epub 2007/01/25. doi: 10.2146/060104. PubMed PMID: 17244877.

2. Ohsfeldt RL, Gandhi SK, Fox KM, Stacy TA, McKenney JM. Effectiveness and cost-effectiveness of rosuvastatin, atorvastatin, and simvastatin among high-risk patients in usual clinical practice. Am J Manag Care. 2006;12(15 Suppl):S412-23. Epub 2006/11/23. PubMed PMID: 17112329.

3. Zhu JR, Tomlinson B, Ro YM, Sim KH, Lee YT, Sriratanasathavorn C. A randomised study comparing the efficacy and safety of rosuvastatin with atorvastatin for achieving lipid goals in clinical practice in Asian patients at high risk of cardiovascular disease (DISCOVERY-Asia study). Curr Med Res Opin. 2007;23(12):3055-68. Epub 2008/01/16. doi: 10.1185/030079907x242809. PubMed PMID: 18196620.

4. Davidson MH. Differences between clinical trial efficacy and real-world effectiveness. Am J Manag Care. 2006;12(15 Suppl):S405-11. Epub 2006/11/23. PubMed PMID: 17112328.

5. Rosenson RS, Otvos JD, Hsia J. Effects of rosuvastatin and atorvastatin on LDL and HDL particle concentrations in patients with metabolic syndrome: a randomized, double-blind, controlled study. Diabetes Care. 2009;32(6):1087-91. Epub 2009/03/07. doi: 10.2337/dc08-1681. PubMed PMID: 19265025; PubMed Central PMCID: PMCPMC2681027.

6. Toth PP, Foody JM, Tomassini JE, Sajjan SG, Ramey DR, Neff DR, et al. Therapeutic practice patterns related to statin potency and ezetimibe/simvastatin combination therapies in lowering LDL-C in patients with high-risk cardiovascular disease. J Clin Lipidol. 2014;8(1):107-16. Epub 2014/02/18. doi: 10.1016/j.jacl.2013.09.009. PubMed PMID: 24528691.

7. Kumar A, Shariff M, Doshi R. Impact of rosuvastatin versus atorvastatin on coronary atherosclerotic plaque volume - a systematic review and meta-analysis with trial sequential analysis of randomized control trials. Eur J Prev Cardiol. 2019:2047487319868035. Epub 2019/08/07. doi: 10.1177/2047487319868035. PubMed PMID: 31382809.

8. Qian C, Wei B, Ding J, Wu H, Cai X, Li B, et al. Meta-analysis comparing the effects of rosuvastatin versus atorvastatin on regression of coronary atherosclerotic plaques. Am J Cardiol. 2015;116(10):1521-6. Epub 2015/09/20. doi: 10.1016/j.amjcard.2015.08.010. PubMed PMID: 26385518.

9. Naci H, Brugts JJ, Fleurence R, Tsoi B, Toor H, Ades AE. Comparative benefits of statins in the primary and secondary prevention of major coronary events and all-cause mortality: a network meta-analysis of placebo-controlled and active-comparator trials. Eur J Prev Cardiol. 2013;20(4):641-57. Epub 2013/03/01. doi: 10.1177/2047487313480435. PubMed PMID: 23447425.

---

## [Editor Report · Decision Letter 1]

1 May 2020

Comparing effectiveness of high-dose Atorvastatin and Rosuvastatin among patients undergone Percutaneous Coronary Interventions: a non-concurrent cohort study in India

PONE-D-19-32015R1

Dear Dr. Roy,

We are pleased to inform you that your manuscript has been judged scientifically suitable for publication and will be formally accepted for publication once it complies with all outstanding technical requirements.

With kind regards,

Corstiaan den Uil

Academic Editor

PLOS ONE
---

## [Editor Report · Acceptance letter]

8 May 2020

PONE-D-19-32015R1 

Comparing effectiveness of high-dose Atorvastatin and Rosuvastatin among patients undergone Percutaneous Coronary Interventions: a non-concurrent cohort study in India 

Dear Dr. Roy:

I am pleased to inform you that your manuscript has been deemed suitable for publication in PLOS ONE. Congratulations! Your manuscript is now with our production department. 

With kind regards,

on behalf of

Dr. Corstiaan den Uil 

Academic Editor

PLOS ONE